# Comparative Analysis of Nucleic Acid-Binding Polymers as Potential Anti-Inflammatory Nanocarriers

**DOI:** 10.3390/pharmaceutics16010010

**Published:** 2023-12-20

**Authors:** Divya Bhansali, Tolulope Akinade, Tianyu Li, Yiling Zhong, Feng Liu, Hanyao Huang, Zhaoxu Tu, Elsie A. Devey, Yuefei Zhu, Dane D. Jensen, Kam W. Leong

**Affiliations:** 1Department of Biomedical Engineering, Columbia University, New York, NY 10027, USA; divya.bhansali@columbia.edu (D.B.);; 2Translational Research Center, College of Dentistry, New York University, New York, NY 10010, USA; ddj3@nyu.edu; 3Pain Research Center, New York University, New York, NY 10010, USA; 4Department of Systems Biology, Columbia University, New York, NY 10027, USA

**Keywords:** toll-like receptor, cell-free DNA, DAMP, PAMP, inflammation

## Abstract

Conventionally, nanocarriers are used to regulate the controlled release of therapeutic payloads. Increasingly, they can also be designed to have an intrinsic therapeutic effect. For example, a positively charged nanocarrier can bind damage-associated molecular patterns, inhibiting toll-like receptor (TLR) pathway activation and thus modulating inflammation. These nucleic acid-binding nanomaterials (NABNs), which scavenge pro-inflammatory stimuli, exist in diverse forms, ranging from soluble polymers to nanoparticles and 2D nanosheets. Unlike conventional drugs that primarily address inflammation symptoms, these NABPs target the upstream inflammation initiation pathway by removing the agonists responsible for inflammation. Many NABNs have demonstrated effectiveness in murine models of inflammatory diseases. However, these scavengers have not been systematically studied and compared within a single setting. Herein, we screen a subset of the most potent NABNs to define their relative efficiency in scavenging cell-free nucleic acids and inhibiting various TLR pathways. This study helps interpret existing in vivo results and provides insights into the future design of anti-inflammatory nanocarriers.

## 1. Introduction

Nanomaterials have revolutionized the field of nanomedicine by enabling the development of novel diagnostic and therapeutic tools for detecting and treating diseases. Nanoparticles, for example, can be designed to target specific cells or tissues in the body, allowing for precise drug delivery and minimizing the side effects of treatments [1,2]. The large surface-area-to-volume ratio of nanomaterials provides for the incorporation of multiple functionalities, such as imaging agents and therapeutic payloads. This multifunctionality has led to the development of therapeutic nanomaterials, which can treat diseases through their intrinsic properties [3]. Scavengers are one form of therapeutic nanomaterials that have shown great promise.

Scavengers are immunomodulatory nanomaterials designed with specific properties to remove overproduced molecules and have therapeutic uses for treating inflammation. They come in many forms, from polymers to nanoparticles to 2D nanosheets. Unlike traditional treatments that only address the symptoms of inflammation, scavengers target the cause by eliminating the agonists that cause toll-like receptor (TLR) overexpression. Scavengers are unique in that they can reduce the immune response in a dose-dependent manner, effectively eliminating overactivation without affecting healthy activation. One promising type of scavenger is nucleic acid-binding nanomaterials (NABNs).

NABNs are nanomaterials that recognize danger- and pathogen-associated molecular patterns (DAMPs and PAMPs) that stimulate TLRs, leading to inflammation in chronic disease [4,5,6]. DAMPs and PAMPs have been clinically found to be increased in trauma [7,8,9], inflammatory disease [10,11,12,13], and cancer [14,15,16]. Cell-free nucleic acids are being used for early disease detection, and high levels of DAMPs and PAMPs have been correlated to lower prognoses in many disease states [7,12,13,16,17,18,19,20,21,22,23,24]. NABNs can reduce TLR overactivation by acting as molecular scavengers that bind nucleic acids and other charged molecules, including DAMPs and PAMPs [25]. By neutralizing the ability of nucleic acids to activate TLRs, NABNs can block inflammation in a controlled and localized manner without compromising the normal immune response to nonnucleic acid pathogenic stimulators. We have seen a variety of scavengers effectively mitigate inflammatory states, including sepsis [26,27], obesity, autoimmune disorders [28,29,30,31], kidney injury [32], cancer metastasis [33,34,35], bone loss [36], wound healing [37], acute respiratory distress syndrome [38], and viral infections.

Dozens of NABNs have been shown to be highly effective in inflammatory disease models. However, these scavengers have never been compared against each other. Here, we screen a subset of the most effective NABNs to see how they compare to one another in scavenging cell-free nucleic acids.

## 2. Materials and Methods

Cell Lines. Human TLR-expressing HEK-Blue hTLR3, hTLR4, and hTLR9 cells were purchased from Invitrogen (Waltham, MA, USA). HEK-293T cells were purchased from GenHunter (Nashville, TN, USA). Cells were cultured in DMEM supplemented with 10% FBS. The medium for HEK-Blue hTLR cells was further supplemented with Normocin (100 µg/mL), Blasticidin (100 µg/mL), and Zeocin (100 µg/mL) (Thermo Fisher, Fair Lawn, NJ, USA).

TLR studies. The ability of the scavengers to inhibit TLR activation was assessed using HEK-Blue hTLR3/4/9 cells. 100 µL of cell solution (5 × 10^5^ cells/mL for TLR 3, 4 × 10^5^ cells/mL for TLR 4, and 8 × 10^5^ cells/mL for TLR 9) was seeded into 96-well plates in DMEM. The following agonist concentrations were used to activate HEK-Blue hTLR3, hTLR4, and hTLR9 cells, respectively: 1 µg/mL poly I:C (Thermo Fisher), a dsRNA analog, was used to activate hTLR3 cells; 2.5 ng/mL LPS (Thermo Fisher) was used to activate hTLR4 cells; and 1 µg/mL ODN BW006 (Thermo Fisher) was used to activate hTLR9 cells. Agonists were added to the cells alone or together with the library NABNs at varying concentrations (1, 5, and 10 µg/mL). The cells were incubated for 24 h. Then, 50 µL of the supernatant media was transferred to a new 96-well plate containing 150 µL of QUANTI-Blue solution (Thermo Fisher) and incubated at 37 °C. The absorbance of the secreted alkaline phosphatase released from the HEK-Blue hTLR cells was measured using a microplate reader at 620 nm. The readings were normalized to the average absorbance of the untreated cells.

Cytotoxicity assay. Cytotoxicity of materials was investigated by using a CCK-8 assay (Creative Bioarray, Shirley, NY, USA). Briefly, TLR9 cells were seeded in 96-well plates at 8000 cells/well and were cultured for 24 h at 37 °C, 5% CO_2_. After treatment with different materials for another 24 h, the medium was replaced with DMEM containing 10% CCK-8 (100 μL) and incubated for 1 h. The absorbance of the cck8 reagent was measured using a microplate reader at 492 nm. The readings were normalized to the average absorbance of the untreated cells.

Endocytosis Study. HEK293T cells were plated on 35 mm glass bottom dishes (MatTek, Ashland, MA, USA) pre-coated with poly-D-lysine. After 24 h, cells were transduced with BacMam 2.0 CellLight Early Endosomes-GFP (1 μL/dish) and BacMam 2.0 CellLight Late Endosomes-RFP (1 μL/dish) (Thermo Fisher). Cells were incubated for 48 h, then washed twice with FluoroBrite DMEM. Cells were incubated with 10 µg PAMAM-Chol-BDP or PG3-Cy5. Cells were imaged live at 0 h, 2 h, and 4 h using the Leica SP8 confocal microscope (Wetzlar, Germany) LAS X V2.6 imaging software. Images were quantified using ImageJ (NIH). ROIs were generated using the late endosome or early endosome channels. A threshold of the channel was set using the Minimum algorithm with a range of 25000-65535 RFU. ROIs were then overlaid to the nanoparticle channel and the mean RFU was measured for each ROI. Nanoparticle-containing endosomes were defined as an endosome (ROI) with a mean RFU >2200 and graphed as the percentage of the total number of endosomes.

### Synthesis of Scavengers

PAMAM-G3. PAMAM-G3-NH2 (Sigma-Aldrich, St. Louis, MO, USA).

Polymer 4 [12]. CDI (1.62 g, 10 mmol) was dissolved in 30 mL of dichloromethane (DCM), followed by dropwise addition of DEEA (0.586 g, 5 mmol) dissolved in 5 mL of DCM to the CDI solution and stirred for 24 h at room temperature. The reaction mixture was washed, and the organic phase was collected, dried over anhydrous MgSO4, filtered, and solvent was removed via roto-evaporation, resulting in a pale yellow product (DEEA-CDI). Then, to create Polymer 4, PAMAM-G3 (30 mg, 4.34 μmol) was dissolved in 2 mL of dry DMSO, and DEEA-CDI (59.0 mg, 277.76 μmol) was added dropwise. The mixture was stirred at 40 °C overnight. The resulting polymer was purified with a Sephadex LH-20 column, and organic solvent was removed via rotary evaporation.

Nanoparticle 5 [12]. Nanoparticle 5 was synthesized using similar methods as Polymer 4. First, DEEA-CDI was synthesized. In the next step, DDC-CDI (6.1 mg, 21.7 μmol) was added to PAMAM-G3 (30 mg, 4.34 μmol) dissolved in 2 mL of dry DMSO, and stirred at 40 °C overnight. Then, DEEA-CDI (118.0 mg, 555.52 μmol) was added, and the mixture was stirred at 40 °C for an additional 24 h. The resulting polymer was purified with a Sephadex LH-20 column, and organic solvent was removed via rotary evaporation. To fabricate nanoparticles, 1 mg of the polymer was dissolved in 200 μL of chloroform, followed by the addition of 1 mL of water and sonication for 2 min. Finally, 5 mL of water was added to the mixture, and the excess solvent was removed using a rotary evaporator to obtain nanoparticle 5.

PAMAM-Cholesterol(5) nanoparticles [6,12]. A solution of 15 μmol of PAMAM-G3 in 5 mL of methanol was mixed with 75 μmol of cholesterol chloroformate in 5 mL of chloroform and 300 μmol of N,N-diisopropylethylamine. The resulting mixture was stirred overnight. The mixture was then dialyzed against water to obtain PAMAM-Cholesterol(5). To fabricate PAMAM-Cholesterol(5) nanoparticles, 1 mg of PAMAM-Cholesterol(5) was dissolved in 200 μL of chloroform, followed by the addition of 1 mL of water and sonication for 2 min. Finally, 5 mL of water was added to the mixture, and the excess solvent was removed using a rotary evaporator to obtain PAMAM-Cholesterol(5) nanoparticles.

G3-coated selenium-doped hydroxyapatite [15]. To synthesize SeHANs, a modified liquid–solid solution (LSS) method was used. First, 1.18 g calcium nitrate tetrahydrate was dissolved in water and combined with an organic solution made of 1.5 g of octadecylamine in 1:4 linoleic acid:anhydrous ethanol. Then, a mixture of trisodium phosphate and sodium selenite dissolved in water was gradually added and stirred for 10 min at room temperature. The suspension was then moved to a hydrothermal reactor and underwent a 12 h reaction at 110 °C. The resulting precipitates were washed with anhydrous ethanol and deionized water and collected by centrifugation. To attach PAMAM-G3 onto SeHANs, a mixture containing 1 mg of SeHANs and 10 mg of PAMAM-G3 in 0.2 mL of PBS (pH 7.4) was incubated for 12 h at room temperature with shaking. The resulting G3-coated SeHANs were centrifuged and washed with PBS.

MnO [13]. MnO nanoparticles were produced through hydrothermal synthesis, utilizing tannic acid and manganese acetate tetrahydrate as precursor solutions at 120 °C. Specifically, manganese acetate tetrahydrate and tannic acid were dissolved in Milli-Q water at room temperature, with a manganese acetate to tannic acid mass ratio of 1:2.8. The mixture was then transferred to an autoclave and heated for 2 h at 120 °C. Following this, the reaction solution was allowed to cool naturally to below 50 °C and then centrifuged at 4000 rpm for 15–30 min to eliminate any large-size flocculent materials. The solution was then further purified via dialysis against distilled water.

CaCO_3_-PEI [18]. CaCO_3_-PEI NPs were prepared via chemical precipitation. 1 mL of Tris-HCl buffer (1 mM, pH 7.6) containing 100 mM CaCl_2_ was mixed with 1 mL HEPES saline buffer (50 mM, pH 7.1, NaCl 140 mM) containing 4 g 25 K PEI, 10 g PEG-PPG-PEG block copolymers, and 8 mM Na_2_CO_3_. The mixture was stirred for 6 h at 4 °C. Excess ions, copolymers, and PEI were removed by dialysis.

MG1 Nanosheet [14]. MoS2 powder (0.8 g) was added to a 100 mL Schlenk flask under an argon atmosphere, followed by the addition of 25 mL of n-butyllithium in n-hexane. The mixture was then condensed and refluxed at 60 °C under an argon atmosphere for 48 h. After that, 50 mL of n-hexane was added to the reaction solution and centrifuged at 2000 rpm for 5 min. The precipitate was washed by centrifugation, concentrated by rotary evaporation, and dialyzed in pure water. The resulting MoS2 nanosheets were then reacted with PAMAM-G1 to create MG-1 nanosheets. Briefly, 0.5 mL of 10 mg/mL lipoic acid solution in methanol was added to 10 mL of 1 mg/mL MoS2 solution in water. The mixture was stirred at room temperature for 24 h to produce LA-modified MoS2 (MoS2-LA). Next, 100 mg (0.52 mmol) of EDC was dissolved in MES buffer to form a 0.5 mM EDC solution, which was then added to the MoS2-LA (10 mg) solution. The solution was slowly added dropwise to 1 g of PAMAM-G1. The reaction mixture was stirred at room temperature for 48 h, and then dialyzed against pure water.

PEI-ZIF-8 [27]. An equal volume of Zn(NO_3_)_2_·_6_H_2_O (14.6 mg/mL) dissolved methanol in was rapidly poured into a solution of 2-methylimidazole (32.4 mg/mL) and PEI (6.0 mg/mL) in methanol and stirred at room temperature for 1 h. The particulates were then centrifuged at 8000 rpm for 5 min, and dialyzed against water.

Tannic Acid-Gentamicin Generation 3 [26]. Tannic acid solution (100 mg/mL), Zn (NO_3_)_2_·6H_2_O solution (50 mg/mL), and 0.5 mL of gentamicin solution (100 mg/mL) were continuously added to water. The mixture was stirred for 10 min at room temperature, and then the pH was adjusted to 5.2 using a sodium hydroxide solution. The mixture was stirred for an additional 20 min, centrifuged at 8000 rpm for 5 min, and dialyzed against water.

## 3. Results

The scavengers screened represent a wide variety of effective NABNs that have been previously shown to be effective in inflammatory disease models. We first characterized the candidates based on their size and surface charge (Table 1). The majority of the NABNs screened are cationic, relying on their charge to act as a molecular trap and collect negatively charged nucleic acids. The majority get their charge from nitrogen-rich polymers, such as PAMAM dendrimers, a family of highly branched polyamidoamine dendrimers composed of a central core, multiple layers of branching, and an outer shell of amino groups. Others, however, are anionic, and use the reducing power of tannic acid to scavenge radicals and act as an antioxidant.

The NABNs screened represent a variety of nanomaterials forms. PAMAM-G3, Polymer 4, Nanoparticle 5, and PAMAM-Cholesterol(5) nanoparticles are all polymeric dendrimer-based NABNs. G3-coated selenium-doped hydroxy apatite, CaCO_3_-PEI_,_ and MnO are nanocomposites and nanoparticles with inorganic cores. Tannic Acid—Gentamicin Generation 3 and PEI-modified ZIF-8 represent various forms of metal–organic frameworks (MOFs). The MG1 nanosheets are a monolayer of molybdenum disulfide (MoS_2_) that are sub-10 nm thick and nanoscale in length and act as a “blanket” to cover and capture larger TLR agonists. Together, they allow us to explore a variety of different attributes that affect scavenging (Figure 1).

Once the scavengers were characterized, we tested their ability to reduce inflammation by scavenging various DAMPs and PAMPs in human TLR reporter cells. The reporter cells have engineered versions of the TLRs, which secrete alkaline phosphatase when activated and are detectable by colorimetric assays. This allows for the relative activation of each TLR to be tested using its specific cell line and measuring the relative amount of alkaline phosphatase in the sample.

DAMPs and PAMPs are known to activate specific TLRs and stimulate the release of cytokines. TLRs 3, 4, and 9 were tested. Each TLR responds to a particular form of molecular pathogen involved in inflammatory states, and would allow us to see if scavengers are TLR-specific or universal scavengers. TLR 3 recognizes double-stranded RNA (dsRNA), a common viral PAMP [25]. TLR 4 recognizes bacteria and endogenous ligands, such as oxidized lipids and heat shock proteins, which are released by damaged cells and tissues [25]. TLR 9 recognizes unmethylated CpG motifs, common in bacterial and viral DNA, and self-DNA [25], which can promote immune pathologies with uncontrolled chronic inflammation. Cell-free RNA (Poly I:C), cell wall components of Gram-negative bacteria (lipopolysaccharide), and methylated cell-free DNA (CpG) were used as agonists to test the efficiency of reducing TLRs 3, 4, and 9, respectively. For TLRs 3 and 9, the agonist-to-scavenger ratio was tested at 1:1, 1:5, and 1:10. For TLR 4, higher agonist-to-scavenger ratios were tested (1:400, 1:2000, and 1:4000) due to the strong activation of TLR 4 due to LPS.

At 1:1, Polymer 1, Polymer 4, PAMAM-G3, MG1 nanosheets, PEI-ZIF8, and PAMAM-Cholesterol(5) all had almost complete inhibition of TLR 3 activation (Figure 2A). At 1:5 and 1:10, G3-SeHan, NP5, MnO, and PEI-ZIF-8 showed significant inhibition. TA-Gen3 and CaCO_3_-PEI showed mild inhibition at 1:10 (Figure 2F,K). Looking deeper, the polymeric NABNs performed similarly. PAMAM-G3 was slightly more effective than its modified counterparts, and PAMAM-Cholesterol(5) had slightly more inhibition than PAMAM-Cholesterol(5) at 1:5 and 1:10 ratios (Figure 2B,G,L). When looking at the anionic materials, MnO was effective at 1:5 and 1:10 ratios, while Ta-Gen3 was ineffective at any ratio (Figure 2D,I,N).

At 1:400, MG1 and PAMAM-Cholesterol(5) completely inhibited TLR 4 activation (Figure 3A). At 1:2000 and 1:4000, NP5, PAMAM-G3, MG1, PEI-ZIF-8, and CaCO_3_-PEI showed moderate inhibition (Figure 3F,K). G3-SeHan, Polymer 4, MnO, and Ta-Gen3 showed little to no inhibition at the ratios tested. Interestingly, some forms of PAMAM-G3 modified materials, namely MG1 and PAMAM-Cholesterol(5), were found to be powerful inhibitors of TLR4, while others, such as G3-SeHan and NP5, were not as effective (Figure 3B,C,G,H,L,M). Neither anionic material was found to affect TLR4 activation (Figure 3D,I,N).

At 1:1, PAMAM-G3, MG1 nanosheets, Ta-Gen3, and PEI-ZIF-8 almost completely inhibited TLR 9 activation (Figure 4A). At 1:5 and 1:10, Polymer 4, NP5, TA-Gen3, and Pamam-Cholesterol(5) showed significant inhibition (Figure 4F,K). CaCO_3_-PEI showed mild inhibition at 1:10 (Figure 4O), while MnO showed no inhibition at the ratios tested. All scavengers showed a dose-dependent effect.

The majority of the NABNs showed no effects on the cytotoxicity of cells at any of the concentrations tested, despite their high charges and efficient scavenging abilities. Interestingly, CaCO_3_-PEI seems to increase cell proliferation in a dose-dependent manner, while MnO appears to have mild toxicity at increased concentrations (Figure 5). This may be because MnO, like many metal oxides, has features of an oxidant, which has unresolved risks when used in biomedical applications [39]. For example, MnO reacts with HCl and forms MnCl_2_. Mn(II) ions may intercalate with DNA, which may be a reason for toxicity [40].

We were also interested in how the difference in scavenging ability may be related to the endosomal retention of materials. To assess this, we studied the uptake of soluble PAMAM-G3 labeled with Cy5-NHS vs. that of PAMAM-Cholesterol(5) NPs labeled with the same dye. We found that soluble polymer was only present in less than 30% of endosomes after 2 h and 50% of endosomes at 4 h. In contrast, nanoparticles were located in almost all early (>90%) and late endosomes at both time points (Figure 6). This explains a possible mechanism by which nanomaterials are better able to scavenge for endosomal TLRs 3 and 9 than soluble polymer scavengers, as they are potentially able to actively block activation in the endosomes.

## 4. Discussion

Interestingly, we found that the nanomaterials on our list that are potent inhibitors of TLR 3 are generally also potent inhibitors of TLR 9. Our cationic NABNs all showed potent inhibition of TLRs 3 and 9. A few modified forms of PAMAM-G3, namely PAMAM-Cholesterol(5) nanoparticles and MG1 nanosheets, were found to be universally potent inhibitors. It is interesting to see that some forms of PAMAM-G3-based cationic materials were effective in inhibiting the effects of TLR 4, while others were not. This may be based on their structure and additional components that may increase the affinity for TLR 4 agonists.

We also found that structure played a significant role in TLR scavenging and efficiency. MG1, a 2D nanosheet, was effectively in scavenging agonists of TLRs 3, 4, and 9, whereas G3-SeHAN, a PAMAM-G3 modified nanocomposite with double the cationic charge, effectively scavenged TLR3 but required a higher ratio to inhibit TLR 9 and had no effect on TLR 4.

Our anionic NABNs had unique properties. TA-Gen3 seems to be a specific scavenger for TLR 9, while MnO is a strong scavenger for TLR 3 but weak for TLR 9. Interestingly, neither NABN scavenged TLR4. This suggests that it is not the tannic acid alone that determines the cfNA scavenging potential of these NABNs. Still, scavenging of agonists of TLR 4 relies on cationic materials for the necessary charge interactions.

The change in endosomal retention of different structures further explains why some scavengers have more effect on endosomal TLRs than others. Large NABNs such as G3-SeHan would be able to inhibit agonists upstream, before they are taken up by the cell. Meanwhile, smaller scavengers can act intracellularly. In the case of nanoparticles that have prolonged endosomal retention, we can expect that, in a more complex in vivo environment, their retention may further increase their efficacy in inhibition of endosomal TLR activation for prolonged periods, which leads to the extended anti-inflammatory effects we have seen in other models [26,27,33,34].

Below, we discuss the intricacies of each scavenger tested.

PAMAM-G3: The generation 3 polyamidoamine dendrimer 3 (PG-3) has 32 primary amino groups on its shell and the highest binding affinity with DNA in our previous screening of polycations. It is a soluble polycation and is effective in a number of our previous studies [41], but it has a relatively high cytotoxicity (IC_50_~30–150 mg/mL). Similarly to what we have seen in various models, PAMAM-G3 has a high affinity to scavenging agonists of TLR 3 and TLR 9, even at low NABN:agonist ratios. Interestingly, we also found a moderate inhibition of TLR 4; however, more complex scavengers showed higher affinities to TLR 4 agonists with more effective inhibition in vitro.

Polymer 4: Polymer 4 is a modified version of Polymer 1, which has an added diethylethanolamine (DEEA) group attached to reduce the toxicity of Polymer 1. The toxicity of Polymer 1 is due to its high cationic charge. By modifying PAMAM amino groups with DEEA, it shields some of the positive charges, reducing toxicity while leaving some amino groups unshielded to allow NA binding. Similarly to what was seen in previous studies [33], we found that Polymer 4 is an effective scavenger of TLR 3 and a moderate scavenger of TLR 9. Expectedly, Polymer 4 had a lesser scavenging ability than Polymer 1 in TLR 4 and TLR 9 but an equivalently strong efficiency in the TLR 3 model. Polymer 4 was also found to be less toxic than Polymer 1.

Nanoparticle 5: This is a nanoparticle based on a modified version of Polymer 4. Polymer 4 was further modified with dodecyl (C12) groups to create amphiphilic polymers to form NPs (Appendix A) that can encapsulate hydrophobic small molecules, such as paclitaxel and doxorubicin. These NPs exhibited similar zeta potentials to the Polymer 1 precursor (+55–65 mV). The level of dodecyl group conjugation can be varied to optimize drug loading, and it was found that higher dodecyl groups led to lower TLR inhibition efficiency and lower cytotoxicity. We studied Nanoparticle 5 (5 dodecyl groups), as it has previously shown the highest DNA binding affinity and small size of the family of NPs [33]. Nanoparticle 5 has previously shown high levels of inhibition of NA-induced activation of TLRs 3, 8, and 9, and high levels of cellular uptake and retention compared to dendrimers. Similar to what was observed in previous studies, Nanoparticle 5 was highly effective in inhibition of TLR 9 and moderately effective in inhibition of TLR 3. Interestingly, we also found mild inhibition of TLR 4 at high NABN:agonist ratios.

PAMAM-Cholesterol(5): This is a nanoparticle formed by the conjugation of cholesterol molecules to PG-3 (Appendix A), which can be controlled by up to 11 molecules. This lowers the cytotoxicity of PG-3 but also decreases its binding affinity with DNA. With as few as two cholesterol molecules per PG-3, the PAMAM-Cholesterol(5) becomes amphiphilic and can self-assemble into a micellar structure. It can encapsulate lipophilic drugs such as paclitaxel and doxorubicin. The NP can be modified by varying the number of cholesterols with n = 2, 5, 8, and 11. This will yield information on the effect of lipophilicity and charge density of NABNs in blocking PAMP/DAMP-induced inflammation. Importantly, the versatility of this nanomaterial can help us find the balance of nucleic acid binding affinity, toxicity, drug encapsulation capacity, and drug release kinetics. We studied PAMAM-Cholesterol(5) as it has previously shown the highest DNA binding affinity and has been effective in several in vivo models [33]. Similarly to what we have seen in various models, PAMAM-Cholesterol(5) has a high affinity for scavenging agonists of TLR 3 and TLR 9, even at low NABN:agonist ratios. Interestingly, we also found a strong inhibition of TLR 4, even at low ratios. PAMAM-Cholesterol(5) may serve as a more universal NABN, able to scavenge a wide variety of agonists.

G3-coated selenium-doped hydroxyapatite: This is a selenium-doped hydroxyapatite nanoparticle (SeHans) coated with the cationic dendrimer PAMAM-G3 to form a cationic nanocomposite (Appendix A). This nanocomposite was designed for topically controlling local bone loss by blocking PAMP/DAMP-induced inflammation. It has been shown to alleviate inflammatory alveolar bone loss in models of periodontitis [36]. Similarly to what was observed in periodontitis models, G3-coated selenium-doped hydroxyapatite is an effective inhibitor of TLR 9 activation at moderate and high concentrations. We also found it to be an effective inhibitor of TLR 3 activation, but not effective against TLR 4 activation. Interestingly, unlike other cationic NABNs, the scavenging mechanism of G3-SeHAN is thought to be completely extracellular due to its large size.

MnO: This is a nanoparticle synthesized by reacting manganese acetate with tannic acid at 120 °C. It is colloidally stable in the PBS buffer for days, with a size of around 50 nm (Appendix A). Interestingly, it has a negative zeta potential of −25 mV, and yet it has a strong DNA binding affinity due to the presence of tannic acid. It has proven efficient in inhibiting TLR activation and macrophage recruitment, blocking CpG-/LPS-TLRs interaction and further downregulating the expression of TNF-α and COX-2 mRNA due to its scavenging ability of pro-inflammatory molecules (e.g., cfDNA, cfRNA and LPS). This MnO has proven effective in inhibiting TLR activation in COVID-19 patient samples, and it limits the ability of DAMPs/PAMPs to induce TLR tolerance in monocytes. It has also proven effective in mitigating breast cancer recurrence and metastasis in a murine model [34]. Similarly to what we have seen in various models, MnO is an effective inhibitor of TLR 3 activation, and a moderate inhibitor of TLR 4 activation.

CaCO_3_-PEI: This is a nanoparticle synthesized via chemical precipitation. It is colloidally stable in the PBS buffer with a size of around 30 nm. Its scavenging ability is thought to come from its cationic charge and incorporation of PEI. CaCO3-PEI was found to be a moderate inhibitor of TLRs 3 and 4 and a mild inhibitor of TLR9.

MG1: This is a 2D nanosheet made of MoS_2_ and decorated with PAMAM-G1 (Appendix A). The MoS_2_ core was chosen due to its biocompatibility and biodegradability. The MG1 nanosheet acts as a blanket to scavenge larger extracellular vehicles that soluble polymer scavengers alone cannot efficiently tackle. Interestingly, NPs exhibited a significantly lower charge than many of their cationic counterparts, yet have been found to have a strong extracellular vesicle and DNA binding ability. MG1 has previously been shown to have a significant effect in reducing radiation-based extracellular vesicle proteins in tumors and are able to decrease metastasis in models of murine breast cancer [35]. Similarly to what we have seen in various models, MG1 has a high affinity for scavenging agonists of TLR 3 and TLR 9, even at low NABN:agonist ratios. Interestingly, we also found a strong inhibition of TLR 4, even at low ratios. PAMAM-Cholesterol(5) may serve as a more universal NABN, able to scavenge a wide variety of agonists.

PEI-modified ZIF-8: This is a metal–organic framework (MOF) synthesized by grafting polyethylenimine (PEI) of different molecular weights to the zeolitic imidazolate framework-8 (PEI-g-ZIF) in a simple one-pot process. PEI_1800_- and PEI_25K_-based MOF structures show efficacy in a murine model of severe sepsis. The nanoparticle size of these two MOF structures varies from 50–300 nm. We investigated the smaller sized NP here, which was synthesized with PEI_1800_, to see its effect on inhibition of the PAMP-induced inflammatory pathways.

TA-Gen3: This is a nanoparticulate scavenger comprising tannic acid, Zn^2+^, and gentamicin designed to target multiple mediators of sepsis to improve sepsis treatment. The TA-Gen3 nanoparticle exerts its anti-sepsis activity by (1) binding cfDNA with high affinity and inhibiting cfDNA-induced activation of TLRs and NF-kB signaling; (2) inhibiting macrophage recruitment; (3) scavenging reactive oxygen species (ROS) and reducing ROS-induced DNA damage and cell death; (4) inhibiting nitric oxide production induced by bacterial lipopolysaccharides; and (5) providing potent antibacterial activity due to the presence of gentamicin. The zeta potential of the nanoparticle varies according to the tannic acid and gentamicin content. We chose to study the variation with the best in vivo effects in sepsis models. This NABN design offers the interesting possibility of directly incorporating antiviral drugs that have primary amino groups into the synthesis of this structure. Similarly to what we have seen in sepsis models, we found TA-Gen3 to be an effective inhibitor of TLR 9 and a mild inhibitor of TLR 3.

## 5. Conclusions

To identify optimal nanomaterials with innate anti-inflammatory properties, we screened the candidates which exhibited favorable properties, including low toxicity, high nucleic acid binding affinity, small size, colloidal stability, and versatility with tunable physicochemical properties, and which have performed well in inflammatory disease models. Our results reveal a notable correlation between the structural properties of NABNs and their TLR inhibitory capabilities. For instance, cationic NABNs, particularly the PAMAM-G3-based materials such as PAMAM-Cholesterol(5) nanoparticles and MG1 nanosheets, demonstrated universal potency in inhibiting TLRs, particularly TLRs 3, 4, and 9. Some scavengers, such as TA-Gen3 and PEI-ZIF-8, are specific to reducing a specific TLR activation. The form of the nanomaterial, whether particulate or 2D in nature, did not play a dominant role in its overall scavenging effect. We found that anionic scavengers such as TA-Gen3 and MnO had varying specificities, showing that the activity of tannic acid alone was not the sole driver of their scavenging. Moreover, the investigation into the endosomal retention of these NABNs provided further understanding of their mechanism of action. Larger NABNs, such as G3-SeHAN, inhibit agonists before cellular uptake, while smaller ones act intracellularly. This difference potentially contributes to the prolonged anti-inflammatory effects observed in various in vivo models. In summary, this study sheds light on the intricate relationship between the structural attributes of NABNs and their function as TLR inhibitors. The findings underscore the potential of these NABNs in developing targeted therapies for inflammatory diseases, paving the way for future research to optimize their design for enhanced efficacy and specificity. These findings will also be useful in understanding the specific mechanisms of complex inflammatory diseases such as autoimmune diseases and pain.

## Figures and Tables

**Figure 1 pharmaceutics-16-00010-f001:**
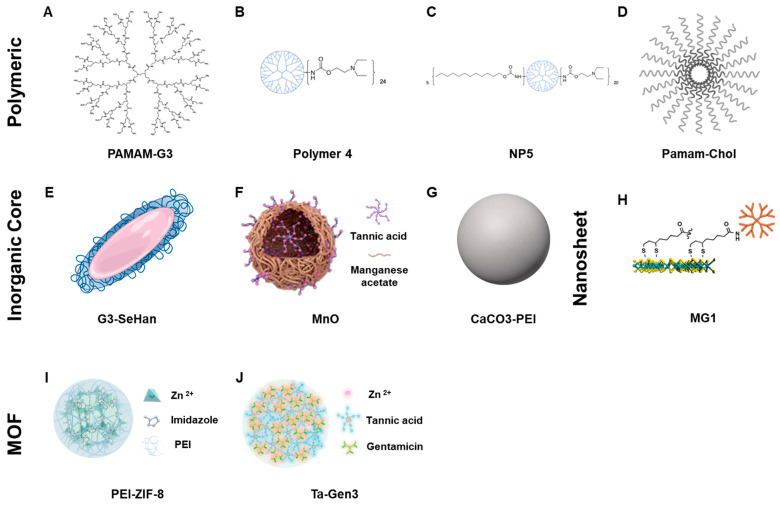
Visualization of scavengers. (**A**–**D**) Polymeric NABNs. (**E**–**G**) NABNs that contain inorganic cores. (**H**) 2D Nanosheet NABN. (**I**,**J**) Metal-oxide framework NABNs.

**Figure 2 pharmaceutics-16-00010-f002:**
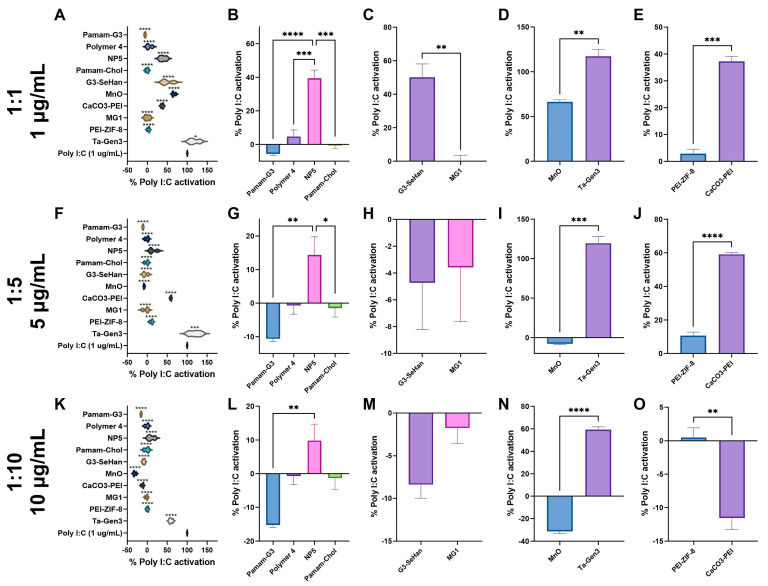
Inhibition of TLR 3 by scavengers. NABNs were tested in various Poly I:C: scavenger ratios, 1:1 (**A**–**E**), 1:5 (**F**–**J**), 1:10 (**K**,**L**) to determine relative inhibition ability of TLR3. Polymeric materials (**B**,**G**,**L**), novel structures (**C**,**H**,**M**), anionic materials (**D**,**I**,**N**), and PEI-based (**E**,**J**,**O**) are compared. * *p* < 0.05, ** *p* < 0.01, *** *p* < 0.001, **** *p* < 0.0001.

**Figure 3 pharmaceutics-16-00010-f003:**
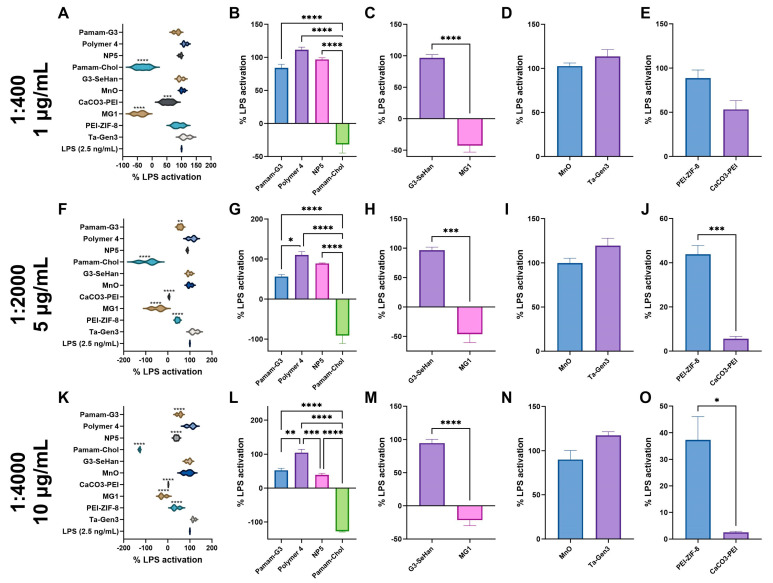
Inhibition of TLR 4 by scavengers. NABNs were tested in various LPS: scavenger ratios, 1:400 (**A**–**E**), 1:2000 (**F**–**J**), 1:4000 (**K**,**L**) to determine relative inhibition ability of TLR4. Polymeric materials (**B**,**G**,**L**), novel structures (**C**,**H**,**M**), anionic materials (**D**,**I**,**N**), and PEI-based (**E**,**J**,**O**) are compared. * *p* < 0.05, ** *p* < 0.01, *** *p* < 0.001, **** *p* < 0.0001.

**Figure 4 pharmaceutics-16-00010-f004:**
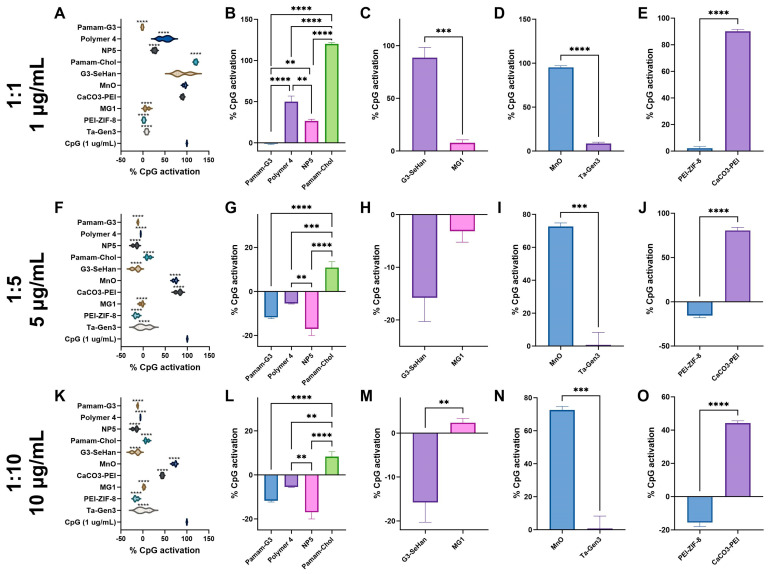
Inhibition of TLR 9 by scavengers. NABNs were tested in various CpG: scavenger ratios, 1:1 (**A**–**E**), 1:5 (**F**–**J**), 1:10 (**K**,**L**) to determine relative inhibition ability of TLR9. Polymeric materials (**B**,**G**,**L**), novel structures (**C**,**H**,**M**), anionic materials (**D**,**I**,**N**), and PEI-based (**E**,**J**,**O**) are compared. ** *p* < 0.01, *** *p* < 0.001, **** *p* < 0.0001.

**Figure 5 pharmaceutics-16-00010-f005:**
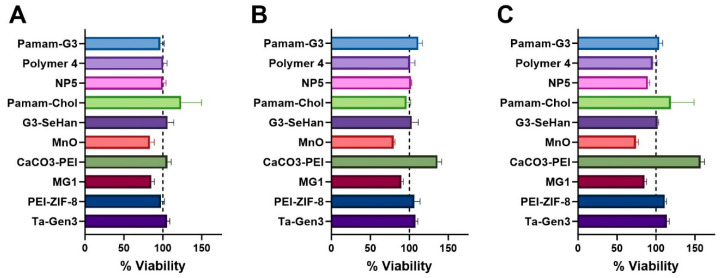
Cytotoxicity of scavengers at 1 µg/mL (**A**), 5 µg/mL (**B**), and 10 µg/mL (**C**) concentrations which correspond to the different ratios tested in inhibition assays.

**Figure 6 pharmaceutics-16-00010-f006:**
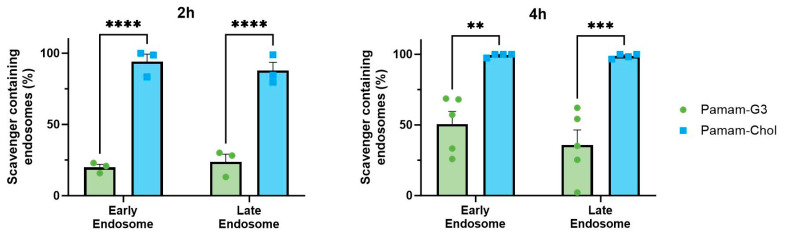
Endosomal uptake and retention of soluble NABN Pamam-G3 vs PAMAM-Cholesterol(5) NP at 2 and 4 h. ** *p* < 0.01, *** *p* < 0.001, **** *p* < 0.0001.

**Table 1 pharmaceutics-16-00010-t001:** Characterization of scavengers.

Name	Charge	Size
PAMAM-G3	**Cationic**	~3.6 nm
Polymer 4	**Cationic:** +25.3 ± 2.8 mV	~3.9 nm
NP5	**Cationic:** +58.2 ± 4.8 mV	141.8 ± 3.3 nm
Pamam-Chol	**Cationic:** +71.5 ± 2.0 mV	154.4 ± 4.7 nm
G3-SeHan	**Cationic:** +26.8 ± 1.9 mV	981.6 ± 48.5 nm
MnO	**Anionic:** −24.1 ± 2.0 mV	34.7 ± 11.1 nm
CaCO_3_-PEI	**Cationic:** +20 mV	~30 nm
MG1	**Cationic:** +10.5 ± 0.5 mV	190.4 ± 7.5 nm
PEI-ZIF-8	**Cationic:** +41.1 ± 1.5 mV	50.5 ± 0.6 nm
Ta-Gen3	**Anionic:** −4.92 ± 0.3 mV	222.6 ± 8.4 nm

## Data Availability

Data are contained within the article and Appendix A.

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
