# Peer review of "Comparative Analysis of Nucleic Acid-Binding Polymers as Potential Anti-Inflammatory Nanocarriers"

_pharmaceutics, 2023, doi:10.3390/pharmaceutics16010010_

Round 1

Reviewer 1 Report

Comments and Suggestions for Authors

The manuscript by Bhansali et al addresses the anti-inflammatory effect of novel polymer nanoparticles TLR3/4/9 signalling pathways. This manuscript is presenting interesting data however the draft has several major points that need to be addressed:

To check the anti-inflammatory effect of candidate polymer nanoparticle HEK Blue hTLR3/4/9 artificial cell model has been used in this study. There are several discrepancies in methodology using this model:

-        There are no justifications why different number of cells is plated per HEK Blue hTLR3/4/9 cell line – different number of cells can affect differentially cell response.

-        The concentration of 500ng/ml of LPS (TLR4 agonist) is extremely high which can lead to TLR4 saturation. HEK Blue hTLR4 cells are extremely sensitive to LPS and max of 100 ng/ml must be used.

-        Treatment approach (combined administration) is limited. If the authors want to check anti-inflammatory pharmacological properties of candidate polymer nanoparticle, the molecule must be administered before TLR agonist, together or after TLR agonist.

Results from HEK Blue hTLR3/4/9 artificial cell model are not expressed properly.

-        The concentrations of candidate molecules must be presented in relevant units (such as mg/ml) not in dilution ratio- this can help researchers to reproduce the results from this study.

-        Data on graphs are not expressed correctly. I recommend authors to show real data per group (for example – untreated, LPS only, LPS + candidate compound etc.) to allow reader to find differences between groups.

In addition to cytotoxicity assay, I suggest authors to use cell viability assay (such as MTT) to make sure that high concentrations of tested molecules cannot affect cell viability.

Comments on the Quality of English Language

.

Reviewer 2 Report

Comments and Suggestions for Authors

Comments

Dear Editor:

In this study, the authors had come up with the idea that the nucleic acid-binding nanomaterials (NABNs), which scavenge pro-inflammatory stimuli, exist in diverse forms, ranging from soluble polymers to nanoparticles and 2D nanosheets. Unlike conventional drugs that primarily address inflammation symptoms, these NABPs target the upstream inflammation initiation pathway by removing the agonists responsible for inflammation. Many NABNs have demonstrated effectiveness in murine models of inflammatory diseases. However, these scavengers have not been systematically studied and compared within a single setting. Herein, the authors screen a subset of the most potent NABNs to define their relative efficiency in scavenging cell-free nucleic acids and inhibiting various TLR pathways. They believe that this study helps interpret existing in vivo results and provides insights into the future design of anti-inflammatory nanocarriers. However, this work needs minor revisions before publication. There are several questions as follow:

In this study,

1.      It would be better to make Table 1 on line 185 a three-line table.

2.      Why does not the PAMAM-G3 in the Table 1 have a specific charge value like other nanomaterials?

3.      Can you explain the meaning of Polymer 1 in line 220 and SS, HPG in line 222? These materials were not mentioned in the previous article.

4.      Why are the CaCO3-PEI and PEI-ZIF-8 in Figure 2, 3, 4 not further compared and analyzed like the other three group?

5.      Why not make further comparisons in animal models?

Reviewer 3 Report

Comments and Suggestions for Authors

Article by Bhansali and others entitled" Comparative analysis of nucleic acid binding polymers as potential anti-inflammatory nano carriers". is scientifically designed experiments. However, final acceptance depends on the clarifications on the following suggestions and/or corrections.

1) More informations is required on the molecular cross-talk between DAMPs and PAMPs as TLRs are know mediators in the molecular signaling pathways. however, other inflammatory pathways informations which cooperatively interact with TLRs should be mentioned in the introductions, which will gives audience on research theme focus about manuscript.

2) Materails methods what is 5X104 cells per well, whats the area of the well, why can't authors mention the international standard of expressions like x cells/ml of medium or x cells/square cm.

3) What is the reference of CCK-assay to assess the cytotoxicity, if authors use commercial kit, they should mention in the parenthesis.

4) Further characterization of NP is required like, electron microscopy, TGA, viscosity, SEM-EDAX.

5). Whats is the bond angle shift when Zn, Tanic acid, Mo and Mn used to synthesis the NP with drugs. Whats is the drug releasing profile.

6) what is the meaning of line # 355, Bone Boss ?

6) Conclusion is feeble must be improved. One important point from the present investigation is that Nanoparticle charge plays important role in TLR activations, any hypothetical or acceptable speculations on charge and activity.

Round 2

Reviewer 1 Report

Comments and Suggestions for Authors

The manuscript by Bhansali et al addresses the anti-inflammatory effect of novel polymer nanoparticles TLR3/4/9 signaling pathways. This manuscript is presenting interesting data however the draft has couple of points that need to be addressed:

To check the anti-inflammatory effect of candidate polymer nanoparticles HEK Blue hTLR3/4/9 artificial cell model has been used in this study. There is discrepancy in methodology using this model:

-        There are no justifications why different number of cells is plated per HEK Blue hTLR3/4/9 cell line – different number of cells can affect the cell response.

Results from HEK Blue hTLR3/4/9 artificial cell model are not expressed properly.

-        Data on graphs are not expressed correctly. I highly recommend authors to show real data (including STDV) per group: for example – Control (not %), LPS only (100%), LPS + candidate compound etc., to allow reader to see differences between groups.